# Pan-Proteomic Analysis and Elucidation of Protein Abundance among the Closely Related *Brucella* Species, *Brucella abortus* and *Brucella melitensis*

**DOI:** 10.3390/biom10060836

**Published:** 2020-05-30

**Authors:** Jayaseelan Murugaiyan, Murat Eravci, Christoph Weise, Uwe Roesler, Lisa D. Sprague, Heinrich Neubauer, Gamal Wareth

**Affiliations:** 1Institute of Animal Hygiene and Environmental Health, Freie Universität Berlin, Centre for Infectious Medicine, Robert-von-Ostertag-Str. 7-13, 14163 Berlin, Germany; uwe.roesler@fu-berlin.de; 2Department of Biotechnology, SRM University AP, Andhra Pradesh, Neerukonda, Mangalagiri, Guntur 522 502, India; 3Institute of Chemistry and Biochemistry, Freie Universität Berlin, Thielallee 63, 14195 Berlin, Germany; murat.eravci@gmx.de (M.E.); chris.weise@biochemie.fu-berlin.de (C.W.); 4Sussex Neuroscience, School of Life Sciences, University of Sussex, Falmer, Brighton BN1 9QG, UK; 5Friedrich-Loeffler-Institute, Institute of Bacterial Infections and Zoonoses, Naumburger Str. 96a, 07743 Jena, Germany; lisa.sprague@fli.de (L.D.S.); heinrich.neubauer@fli.de (H.N.); gamal.wareth@fli.de (G.W.); 6Faculty of Veterinary Medicine, Benha University, Moshtohor, Toukh 13736, Egypt

**Keywords:** pan-proteomics, *Brucella abortus*, *Brucella melitensis*, label-free quantitative analysis, LC–ESI–MS/MS

## Abstract

Brucellosis is a zoonotic infection caused by bacteria of the genus *Brucella*. The species, *B. abortus* and *B. melitensis*, major causative agents of human brucellosis, share remarkably similar genomes, but they differ in their natural hosts, phenotype, antigenic, immunogenic, proteomic and metabolomic properties. In the present study, label-free quantitative proteomic analysis was applied to investigate protein expression level differences. Type strains and field strains were each cultured six times, cells were harvested at a midlogarithmic growth phase and proteins were extracted. Following trypsin digestion, the peptides were desalted, separated by reverse-phase nanoLC, ionized using electrospray ionization and transferred into an linear trap quadrapole (LTQ) Orbitrap Velos mass spectrometer to record full scan MS spectra (*m/z* 300–1700) and tandem mass spectrometry (MS/MS) spectra of the 20 most intense ions. Database matching with the reference proteomes resulted in the identification of 826 proteins. The Cluster of Gene Ontologies of the identified proteins revealed differences in bimolecular transport and protein synthesis mechanisms between these two strains. Among several other proteins, antifreeze proteins, Omp10, superoxide dismutase and 30S ribosomal protein S14 were predicted as potential virulence factors among the proteins differentially expressed. All mass spectrometry data are available via ProteomeXchange with identifier PXD006348.

## 1. Introduction

*Brucella* represents a Gram-negative bacterial genus of the α-2 subgroup of *Proteobacteria.* Brucellae are highly adapted to their intracellular lifestyle and are the causative agents of human and animal brucellosis (“undulant fever”, “Malta fever”, “Mediterranean fever” or “Bang’s disease”) [1]. They are highly infective and 10–100 bacteria cause human infection [2,3]. The genus *Brucella* currently includes 12 accepted species that have been named according to their host specificity. To date, the mechanism behind the host specificity is not clear [4]. The classification of *Brucella* species is under debate due to the reported high degree of homology found in DNA-DNA hybridization studies. This results in the proposal that the genus *Brucella* is a single genomospecies and the species are only biovars of *B. melitensis* [5,6]. It is also believed that *B. abortus* and *B. melitensis* share a common ancestor that evolved from *B. suis* [7]. *B. abortus* (preferred host: cattle and other bovidae), *B. melitensis* (small ruminants such as goats and sheep) and *B. suis* (pigs) cause human brucellosis. The Centers for Disease Control and Prevention has listed Brucella species on the Federal Select Agent Program/Select Agents and Toxins List (https://www.selectagents.gov/SelectAgentsandToxinsList.html, assessed in March 2020) and on the Emergency Preparedness and Response Bioterrorism Agents/Diseases List (https://emergency.cdc.gov/agent/agentlist.asp, assessed in March 2020) as prepared by the National Center for Emerging and Zoonotic Infectious Diseases.

*B. melitensis* and *B. abortus* have striking similarities, i.e., two chromosomes, comparable gene sequence, organization and structure [7,8,9]. These two species are the most often sequenced species of *Brucella* with 256 genome assemblies of *B. abortus* and 254 genome assemblies of *B. melitensis* listed at the NCBI database (March 2020) [8,9]. The genome size of the *B. abortus* type strain 544 was reported to be 3,289,405 bp and was predicted to possess 3319 genes among which 3259 were protein-coding genes and 60 RNA genes [10]. The genome of *B. melitensis* strain 16M encompasses 3,294,935 bp distributed over two circular chromosomes and contains 3197 open reading frames (ORFs), potentially available for expression [11]. Comparison of the *B. melitensis* 16M genome with strains of five other *Brucella* species revealed alterations in ORFs and species-specific conservation in terms of genetic content deletion or missing genome islands [12]. The host specificity might be caused by the species-specific gene inactivation/activation that influences transcriptional regulators and outer membrane proteins [7,13,14]. Previous studies indicated that these two strains vary in their antigenicity, immunologic and genetic properties [15,16,17]. There is no clear information in the literature designating whether live attenuated *B. abortus* and *B. melitens* is strains provide cross-protection among bovines and small ruminants respectively [18,19]. These two species display differences in phenotype, e.g., dye sensitivity, CO_2_ requirement or H_2_S production, which are used for diagnosis and biotyping [4,20,21,22]. Differences in immuno-dominant proteins of field isolates were demonstrated using sera collected from naturally infected animals [23,24]. Although it is broadly accepted in the field that the Type IV secretion system (T4SS) is in some way tied to virulence in *Brucella* species, *Brucella* generally lacks classical virulence factors and in order to explain its virulence and host specificity, a better understanding at the proteome and metabolome level is needed. Proteomic analyses of various strains of *B. abortus* and *B. melitensis* have been reported [25,26,27,28,29,30,31,32,33,34], suffering from limitations in terms of technology and database coverage. In the present study, the label-free quantitative proteomic analysis includes the reference strains as well as strains isolated from infected animals analyzed in an earlier study [23]. The focus of this study was to investigate proteome level differences between *B. abortus* and *B. melitensis* cultured under laboratory conditions and relate them to possible factors involved in mechanisms of virulence and differences in host specificity.

## 2. Materials and Methods

### 2.1. Brucella Culture

*Brucella* type strains and field isolates as listed in Table 1 were from the culture collection of the Friedrich-Loeffler-Institut (FLI), Federal Research Institute for Animal Health, Institute of Bacterial Infections and Zoonoses (IBIZ), Jena, Germany. Each strain was independently cultivated 6 times in 50 mL of Tryptic Soy Broth at 37 °C in the presence of 5% CO_2_ with shaking until the CFU was around 5 × 10^8^ cells/mL. The cells were harvested by centrifugation at 11290×
*g* for 5 min and after washing twice with phosphate buffer saline, the cells were inactivated and fixed by reconstituting the cell pellets with 300 µL of high performance liquid chromatography (HPLC) grade distilled water and 900 µL of absolute ethanol.

### 2.2. Whole-Cell Protein Extraction

In order to extract proteins from the ethanol-fixed cells, cells were centrifuged at 11,290× *g* for 2 min, the supernatant was discarded and the resultant cell pellets were air-dried for 20 min to remove ethanol traces. The cell precipitate was then reconstituted in 250 µL of lysis buffer (20 mM HEPES, pH 7.4), sonicated on ice for 1 min (duty cycle: 1.0, amplitude: 100%, UP100H; Hielscher Ultrasound Technology, Teltow, Germany), centrifuged at 11,290× *g* for 5 min at 4 °C and the supernatant collected. The protein content was measured using a modified Bradford’s method (Biorad, Munich, Germany). The values obtained were checked for consistency by Sodium Dodecyl Sulfate PolyAcrylamide Gel Electrophoresis (SDS-PAGE) [35]. A volume of the whole-cell extract containing 10 µg of protein was subjected to acetone precipitation, reconstituted in 10 μL sample loading buffer, heated for 5 min at 60 °C and subjected to gel electrophoresis (4% acrylamide concentration in the stacking and 12% acrylamide concentration in the separating gel); the protein bands were visualized using Coomassie staining [36]. 

### 2.3. In Solution Trypsin Digestion

The protein extract containing 10 µg of protein was subjected to acetone precipitation and trypsin digestion as described elsewhere [37]. In brief, following acetone precipitation, the precipitate was reconstituted with 10 µL of denaturation buffer (6 M urea/2 M thiourea in 10 mM HEPES, pH 8.0). All steps of in-solution trypsin digestion were carried out at room temperature. The reduction was carried out for 30 min by adding 0.2 µL of 10 mM dithiothreitol in 50 mM of ammonium bicarbonate (ABC). Subsequently, alkylation was performed for 30 min by adding 0.2 µL of 55 mM iodoacetamide in 50 mM ABC. Then, 0.4 µL of LysC (0.5 µg/µL; Wako, Neuss, Germany) in ABC solution was added and incubated overnight at room temperature. Next, 75 µL of ABC were added to decrease the urea concentration to <2 M to enable trypsin digestion. Trypsin digestion was carried out overnight at 37 °C after adding 0.4 µL of 0.5 µg/µL trypsin in 50 mM ABC and the reaction was arrested by adding 100 µL of 5% acetonitrile in 3% trifluoroacetic acid. 

### 2.4. Liquid Chromatography–Electrospray Ionization–Tandem Mass Spectrometry (LC–ESI–MS/MS)

The trypsin-digested peptides were first desalted by solid-phase extraction, using the stage-tip procedure [38]. Nano liquid chromatography–tandem mass spectrometry (LC–MS/MS) analysis was carried out using a Dionex Ultimate 3000 nanoLC system (Dionex, Germering, Germany) coupled with an LTQ Orbitrap Velos mass spectrometer (Thermo Fisher Scientific, Bremen, Germany), operated in data-dependent acquisition mode with the Xcalibur software (version 21.0.1140, Thermo Fisher Scientific). The nanoLC system was used to load the peptides in 0.1% formic acid onto a C18 PepMap trap column (75 µm ID × 2 cm, Dionex). Then, separation was achieved with a 5–60% acetonitrile gradient (90 min) with 0.1% formic acid at a flow rate of 350 nL/min through a 25 cm fritless C18 microcolumn packed inhouse with ReproSil-Pur C18-AQ 3 μm resin (Dr. Maisch GmbH, Entringen, Germany). Online electrospray ionization with an electrospray voltage of 2 kV was used for direct ionization of the eluted peptides. The ions were then transferred into an LTQ Orbitrap Velos operated in the positive mode to record full scan MS spectra (from *m/z* 300–1700) at a resolution of R = 60,000 followed by isolation and fragmentation of the 20 most intense ions by collision-induced dissociation.

### 2.5. Protein Identification 

All raw MS files were combined and processed with the MaxQuant software (version. 1.6.0.16/Max-Planck-Institute of Biochemistry, Martinsried, Germany) [39,40]. The following parameters were set for protein identification: minimum required peptide length, seven amino acids, enzymes, LysC and trypsin, both enzymes with two missed cleavages, fixed modification, cysteine carbamidomethylation and variable modifications, oxidation of methionine and protein N-terminal acetylation. The initial precursor and fragment ion maximum mass deviations were set to 7 ppm and 0.5 Da, respectively, for the search against forward and backward protein sequences of a combined *Brucella* database (*B. abortus* 2308 and *B. melitensis* M28) downloaded from the UniProt Knowledgebase. The target-decoy-based false discovery rate (FDR) for peptide and protein identification was set to 0.01 to ensure that the proteins identified with the lowest score had a probability of ≤1% of being a false identification. The most frequently observed laboratory contaminants were eliminated from the list of identified proteins and the proteins with at least one peptide unique to the protein sequence were considered as valid identifications. MS-based quantification of proteins was performed using the label-free quantification algorithm of the MaxQuant software package [41,42].

### 2.6. Data Analysis

The data analysis was carried out using the freely available software Perseus (version 1.4.1.3; Max-Planck-Institute of Biochemistry, Martinsried, Germany), after importing the label-free quantification (LFQ) intensities of the proteins from the MaxQuant analysis. The intensities were first transformed to a logarithmic scale with base two and the missing values were replaced (imputated) with the value of the lowest observed value in the dataset. Statistical analysis was carried out using a two-way Student *t*-test, error correction (*p* < 0.05) and FDR correction of the alpha error was carried out by the method of Benjamini–Hochberg [43]. The comparisons between the four different datasets (type and reference strains of each species) were carried out in different pairs. Heat map and hierarchical clustering of proteins (Euclidean distance and linkage) were calculated using *z*-score normalized and logarithmized intensities of identified proteins. For further visualization, volcano plots and principle component analysis (PCA) were performed. All the proteins that showed a fold-change of at least 1.5 and met *p* > 0.05 were considered differentially expressed.

The mass spectrometry data have been deposited to the ProteomeXchange Consortium via the PRIDE partner repository [44,45], with the dataset identifier PXD006348.

### 2.7. Functional Categorization and Pathways Analysis

The UniProt FASTA files of protein sequences were analyzed using http://eggnogdb.embl.de (assessed in February 2017) to achieve the functional annotation of the identified proteins in terms of clusters of orthologous group (COG) [46]. The canonical pathways were also analyzed using the Database for Annotation, Visualization and Integrated Discovery (DAVID) tool [47,48].

### 2.8. Screening for Virulence-Associated Proteins

Protein sequences downloaded from Uniprot in FASTA format were used to predict the virulence nature using the VirulentPred online analysis tool (http://bioinfo.icgeb.res.in/virulent/ assessed in Sep. 2019) [49]. 

### 2.9. Mass Spectrometry Data

The mass spectrometry proteomics data have been deposited to the ProteomeXchange Consortium [50] via the PRIDE partner repository with the dataset identifier PXD006348.

## 3. Results and Discussion

In the present study, the reference strains *B. abortus* (strain 544, ATCC 23448) and *B. melitensis* (strain 16M, ATCC 23456) [10,11,26] were chosen in an attempt to understand the proteomic differences between strains of closely related *Brucella* species. The strains isolated from infected animals that were previously used to demonstrate the existence of protein expression level differences [23] were also included for comprehension. Differences at the proteome level may also underlie differences in their phenotypes, pathogenicity and host specificity [22,51,52]. The vaccine strain Rev. 1 and the laboratory strain B115 of *B. melitensis* had comparable two-dimensional gel electrophoresis (2DE) protein patterns. However, the reference strain 16M displayed 50% fewer protein spots [53]. Strains of the same species possessing homologous genomes and displaying comparable phenotype or biochemical reactions [54] also displayed different proteomes in *B. abortus* strains, i.e., the virulent strain 2308 and the vaccine strain S19 [55]. Earlier genome-based suppressive subtractive hybridization studies had identified species-specific deletions which 2DE-based investigations could not confirm [27,28,56] due to the limitations in available protein identification coverage and of protein entries in the database. Factors such as heat, oxidative and acidic pH stress and culture media have influenced the 2DE based protein coverage of *B. abortus* and *B. melitensis* [32,57]. Application of LC–MS has enhanced the protein coverage as demonstrated in the case of reference strain *B. abortus* 2308 to create a dataset of 621 proteins among which 300 were not reported earlier and five were attributed to pseudogenes [29]. LC–MS-based quantitative proteomic comparison of the outer membrane fraction of virulent and avirulent strains of *B. abortus* results in the fact that *Brucella* virulence is based on extensive cell envelope-based modifications [26,30,55]. Therefore, the present study involving LC–MS-based quantitative proteomic analysis of whole-cell protein extracts was initiated to identify protein expression level differences between these two closely related bacterial species.

### 3.1. Brucella Whole-Cell Protein Extraction

Preliminary analyses using SDS-PAGE separation (Figure 1) revealed that both *B. abortus* and *B. melitensis* express very similar sets of proteins with regard to protein pattern and band intensities as well as the occurrence of few differences such as a distinct band around 35 kDa (*B. abortus*) and below 20 kDa (*B. melitensis*). The type strains and field strains displayed comparable bands but with varying intensities.

### 3.2. Databases and Protein Sequences

Protein identification by matching mass spectra against a database of known sequences is an important tool in proteomics. Despite the availability of a list of differentially expressed proteins and the influences of the physicochemical environment, missing or unknown sequences remain a limiting factor. UniProt listed as many as 458 proteome entries for *Brucella*, however, 329 entries of proteomic data were redundant and moved to the UniProt Archive (UniParc) database. The remaining nonredundant database represents 10 *Brucella* species and remains active in the UniProt KB database. UniProt introduced the new term “pan proteome” to describe the entire set of proteins thought to be expressed by a group of highly related organisms, e.g., multiple strains of a species. The pan database entries also included all sequences within a taxonomical group as well as unique sequences not found in the reference proteome [58]. Consequently, 25 proteomes representing 10 species of *Brucella* were used to create the *B. abortus* 2308 pan proteome and the *B. abortus* (strain 2308) proteome remained the reference proteome. An analysis of protein IDs of the pan proteome and five strains of *Brucella* sp. based on the online software tool InteractiVenn [59] revealed that the complete list of the *Brucella* reference proteome (except three entries) forms 47% of the pan proteome (Figure 2). Moreover, the bulk of protein entries in the pan proteome were from *B. abortus, B. melitensis*, *B. suis, B. ovis* and *B. vulpis,* respectively. The *Brucella* pan proteome contains 7266 protein entries. Their existence was mostly predicted (77%) or inferred from homology (21.6%) whereas evidence at the protein level (1%) and transcript level (0.1%) was scarce. The manual curation of protein entries was reported for 528 protein entries which correspond to approximately 7.3% of *Brucella*-specific protein entries. For MS-based proteome analysis of two species, pan proteome or inclusion of all 10 *Brucella* proteomes for protein identification might be inconvenient due to the difference in the entry IDs for each species. Therefore, for the sake of effective protein identification, the following two proteomes were combined: (1) *B. abortus* (strain 2308) with a protein count of 3023, distributed on chromosome I (1991) and II (1034, proteome UP000002719, organism ID: 359391. update: 11 February 2017) and (2) *B. melitensis* biotype 1 (strain 16M/ATCC 23456/NCTC 10094) with a protein count of 3178 distributed on chromosome I (2049) and II (1131, proteome UP000008511, organism ID: 224914. update: 04 February 2017). The latter was redundant and replaced with *B. melitensis* with a protein count of 3123 distributed on chromosome I (1075) and II (2048, proteome UP000290786, organism ID: 29459. update: 01 December 2019, accessed on March 2020). This choice was based on the designation of *B. abortus* (strain 2308) as reference proteome and the highest number of protein entries among the three available *B. melitensis* proteomes. 

### 3.3. Protein Identification

A MaxQuant-Andromeda-based search against the combined database of *B. abortus* (strain 2308) and *B. melitensis* biotype 1 (strain 16M) resulted in the identification of 1202 proteins with at least one unique peptide specific for a protein. A total of 826 proteins were identified, after applying the filter that label-free quantification intensity (LFQ) of a protein was present in at-least four out of six replicates in each sample dataset and after removal of proteins matched toreverse sequences and those proteins identified “by site” (Appendix A). Among these identified proteins, 478 protein IDs belonged to the *B. abortus* reference proteome and the remaining 348 protein IDs could be allocated to the *B. melitensis* proteome. The distribution of the identified proteins with respect to the chromosomes was 360 on chromosome I (43.6%) and 118 on chromosome II (14%) of *B. abortus* and 161 proteins expressed on chromosome I (19.5%) and 56 on chromosome II (6.8%) of *B. melitensis.* The remaining 131 proteins (15.7%) belonged to *B. melitensis* unassembled WGS sequences, which are also part of the *B. abortus* 2308 pan proteome. Later, a database update resulted in the addition of one protein in chromosome I of *B. abortus* (strain 2308, update 05 December 2016). The proteome UP000008511 was moved to UniParc as it was identified as redundant (update 18 November 2016) and the majority of its protein entries were found to be matching with proteome UP000000419 (update 9 October 2016). As a result, 131 protein entries (belonging to proteome UP000008511: *B. melitensis* biotype 1) of the 826 identified proteins were found redundant (marked as removed in Appendix A) or obsolete and deposited at the UniParc database. Consequently, the remaining 695 identified proteins were considered for further analysis. 

### 3.4. Comparative Proteomics of B. abortus and B. melitensis 

Visualization through unsupervised hierarchical clustering of the proteomic data recapitulated the similarities between strains and species (Figure 3). All six replicates of each isolate clustered together and the species displayed a clear clustering. For comparative proteomics analysis, pairwise comparisons were carried out in six categories as follows, 

Category I: *B. melitensis* 16M vs. *B. abortus* 544 (M vs. A); Category II: *B. melitensis* C vs. *B. abortus* T (M2 vs. A2);Category III: *B. melitensis* C vs. *B. abortus* 544 (M2 vs. A);Category IV: *B. melitensis* 16M vs. *B. abortus* T (M vs. A2);Category V: *B. abortus* T vs. *B. abortus* 544 (A2 vs. A); Category VI: *B. melitensis* C vs. *B. melitensis* 16M (M2 vs. M).

As shown in Figure 3, the volcano plot displays the negative log_10_
*t*-test *p*-value over the log_2_ fold-change. Proteins with *p*-values above the dotted line (*p* < 0.05) were considered to be differentially expressed between the two groups. The left side of the plot represents the downregulated proteins and the right side represents the upregulated proteins. Table 2 lists the differentially expressed proteins in each of the above-described categories. The proteins identified as significantly regulated in at least three of the four comparison categories (I–IV) were considered to play a crucial role. As a result, 109 and 104 proteins were identified as up- or downregulated in *B. melitensis* when compared to *B. abortus* (Appendix A). 

### 3.5. Geno Ontology and Clusters of Orthologous Groups 

UniProt was used to cluster the proteins in accordance with their Gene Ontologies (GO) for understanding the functional role of the identified proteins. The GO mapping was possible only for those proteins that remained active at the UniProt database until the analysis. GO results presented in Figure 4 include 695 identified proteins among which 66 and 94 were up- or downregulated, respectively, in *B. melitensis* compared to *B. abortus,* while Clusters of Orthologous Groups (COGs) are indicated for all 826 identified proteins including 131 redundant proteins. Catalytic activity, binding properties, transporter and antioxidant activity, as well as ribosomal structural constitution, were different between these two species. Macromolecular, membrane and cellular components also varied. The major metabolic and cellular processes appeared to be similar between these species. The prediction of COGs revealed the distribution within four functional groups: cellular processes and signaling (D, M, O, T and U), information storage and processing (J, K and L), metabolism (C, E, F, G, H, I, P and G) and poorly characterized (S). Analysis of about 20% of the identified proteins identifies any COG. Based on the COG clustering, membrane proteins and proteins involved in biomolecular transport and protein synthesis mechanism were found to be enhanced in *B. abortus* in comparison to *B. melitensis*. 

### 3.6. Bioinformatics Annotation of Differentially Expressed Proteins

As shown in Table 3, the DAVID analysis of differentially expressed proteins revealed involvement of several Kyoto Encyclopedia of Genes and Genomes (KEGG) pathways. Notably, histidine metabolism appeared to be different in all compared groups. As shown in Figure 5, among the top 10 pathways, the observed differences in the categories ABC transporters, aminoacyl-tRNA biosynthesis and oxidative phosphorylation merit further investigation to clarify, if these pathways influence biochemical diagnosis or host specificity. The role of ABC transporters in intracellular survival and virulence of *Brucella* was demonstrated in *B. ovis* [60]. It was also shown that about 9% of the coding ability of *Brucella* is devoted towards ABC transporters, but differences between these two *Brucella* species were reported [61]. Analysis using DAVID indicated that the proteins downregulated in *B. abortus* in comparison to *B. melitensis* occurred within three biological pathways, each with four of the proteins identified: pyruvate metabolism, histidine metabolism, arginine and proline metabolism and lysine degradation, and tryptophan metabolism. In contrast, the upregulated proteins were present in three other pathways: 11 proteins in metabolic pathways, four proteins in carbon metabolism and three proteins in pyrimidine metabolism. 

### 3.7. Predicted Virulence-Associated Proteins

The prediction of protein pathogenicity was carried out using the freely available online Support Vector Machines (SVM)-based tool VirulentPred [49]. Of the 826 proteins identified, 102 proteins (12%) were predicted to be potentially virulence-associated (Supplementary Table 3) among which 22 proteins were identified as differentially expressed proteins (16 upregulated and six downregulated in *B. abortus* comparison to *B. melitensis*). With the exception of the proteins listed in Table 4, all other differentially expressed proteins predicted as virulence-associated were listed as uncharacterized proteins. The upregulation of potentially virulence-associated proteins and downregulation of ribosomal proteins indicate different degrees of control operations that prepare the bacterial agent for infection [62]. The identified proteins can be explored for further application in diagnostics. 

### 3.8. Field Strains and Host Adaptability

Within the same species, the type and field strains also displayed significant variations in the protein abundances. As listed in Appendix A, the type strain and field strain of *B. abortus* showed differences in the expression of 180 proteins, among which 97 and 83 proteins were up- and downregulated, respectively, in the field strain when compared to that of the type strain. On the other hand, *B. melitensis* displayed differences in 224 proteins, among which 129 and 95 were found to be up- or downregulated in the field strain. Among these differentially expressed proteins, as listed in Table 5, 10 proteins including several binding proteins—appeared to be highly abundant among the type strains of both species, while 19 proteins—mostly belonging to the Type IV secretion system (T4SS)—were highly abundant among the field strains of *B. abortus* and *B. melitensis*. T4SS has been associated with increased host adaptability and described as an essential pathogenicity factor in several pathogens including *Brucella* spp., *Helicobacter pylori*, *Legionella pneumophila* and *Bartonella* spp [71,72]. T4SS influences the intracellular survival in the host [73,74]. Nine proteins identified as upregulated in the *B. abortus* field strain were found to be downregulated in the field strain of *B. melitensis*. On the other hand, seven proteins identified as upregulated in the *B. melitensis* field strain were found to be downregulated in the *B. abortus* field strain. These proteins are worth further analysis as they might play a role in the known host-species specificity and might be useful for designing species-specific diagnostic tools.

## 4. Conclusions

In conclusion, our quantitative proteomic analysis of reference and field-isolated strains of *B. abortus* and *B. melitensis* not surprisingly confirms the existence of proteome level differences between the strains. Besides differences in metabolic pathways, *B. abortus* and *B. melitensis* displayed differences in ABC transporters, which were shown to play a role in intracellular survival and virulence of *Brucella*. Field isolates displayed enhanced abundance in several binding proteins and Type IV secretion systems (T4SS), these have been associated with host adaptability and essential pathogenic factors. *B. abortus* field strain displayed a high abundance of 10 proteins and seven proteins were of high abundance in *B. melitensis.* These proteins might be playing a role in host specificity. With the exception of seven proteins, all other proteins (*n* = 15) identified as potentially virulence-associated were uncharacterized proteins. Problems arise from the existence of multiple redundant reference proteomes for different *Brucella* species. The benefits of the recently introduced pan proteome concept based on protein entries of 10 known *Brucella* species also remain limited, as long as the majority of protein entries at the UniProt database remain unreviewed/curated. Establishing a species-specific proteome would be useful for understanding the host specificity and infection of *Brucella* species. We suggest that in addition to improvements in the reference database, further in-depth proteomic analyses are performed on these strains cocultured with their respective host cell lines. This might lead to an understanding of the mechanism lying behind the described host specificity and pathogenicity.

## Figures and Tables

**Figure 1 biomolecules-10-00836-f001:**
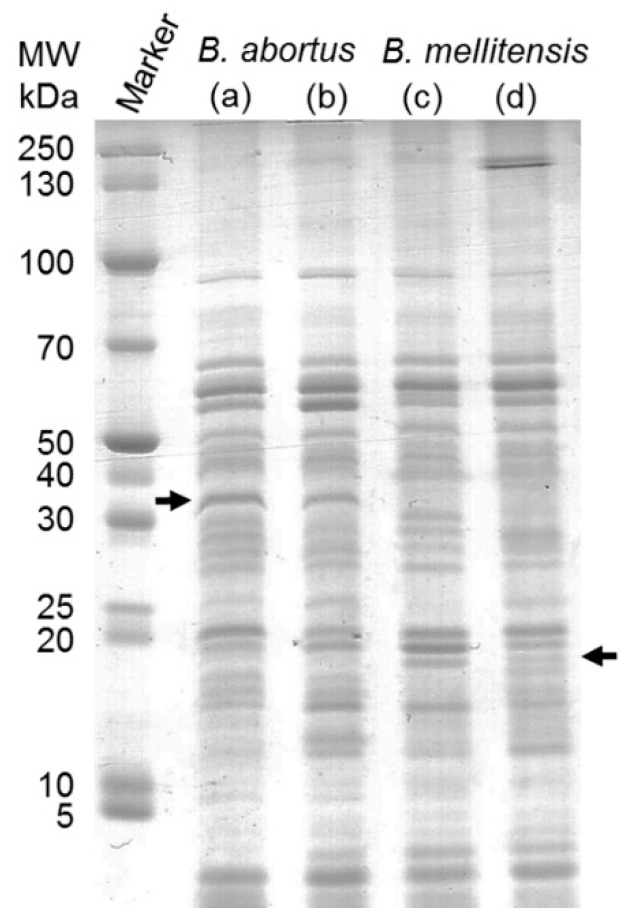
SDS-PAGE separation of whole-cell extract: Lane (**a**) *B. abortus* 544, (**b**) *B. abortus* T (**c**) *B. melitensis 16 M* and (**d**) *B. melitensis* C. The strains display comparable bands. Arrows show distinct bands unique for the isolates with 35 kDa in *B. abortus* and below 20 kDa in *B. melitensis*.

**Figure 2 biomolecules-10-00836-f002:**
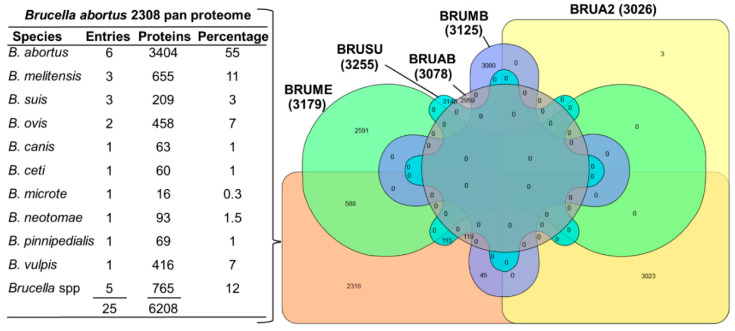
The table lists the *Brucella* species (*n* = 10) included in the *B. abortus* 2308 pan proteome. Entries: number of isolates; proteins: number of protein entries and percentage: indicates percentage found in the *B. abortus* 2308 pan proteome. InteractiVenn diagram of *Brucella abortus* 2308 pan proteome and the entries of five strains (BRUA2—*B. abortus* (strain 2308, yellow), BRUAB—*B. abortus* biovar 1 (strain 9-941, gray), BRUME—*B. melitensis* biotype 1 (strain 16M/ATCC 23456/NCTC 10094, green), BRUMB—*B. melitensis* biotype 2 (strain ATCC 23457, purple) and BRUSU—*B. suis* biovar 1 (strain 1330), blue).

**Figure 3 biomolecules-10-00836-f003:**
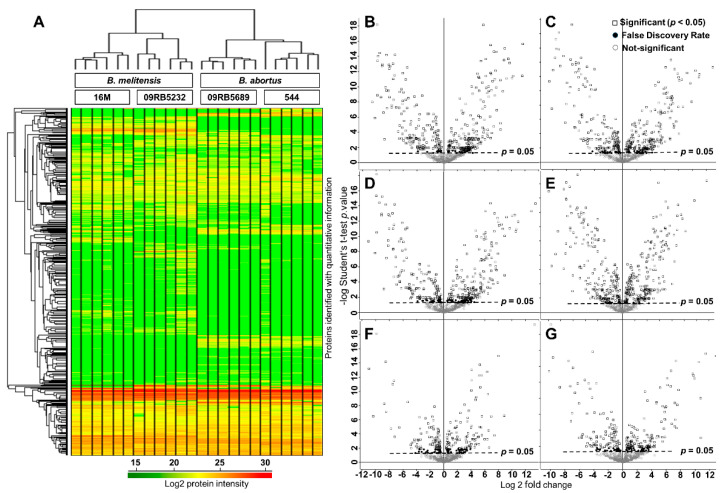
Protein expression profiling using label-free quantitative proteomics analysis. (**A**) Heat map analysis of 826 proteins among the four dataset groups and with six cultural replicates per group. The log_10_ value of the MS signal intensity is shown. Hierarchical clustering of proteins of all samples was performed using *z*-score protein intensities for the proteins with *p* < 0.05 and based on elucidation distance. Columns indicate the samples, and rows indicate the proteins. Protein expression values were log_2_-normalized (label-free quantification (LFQ)) intensities of all proteins quantified across the samples, where red and green indicate high and low intensity, respectively. (**B**–**G**) Volcano plot for category I–VI, respectively. Ratios plotted for log 2-fold-change (*x*-axis) against negative log *p*-value (*y*-axis) of the Student’s *t*-test. Each dot represents a protein.

**Figure 4 biomolecules-10-00836-f004:**
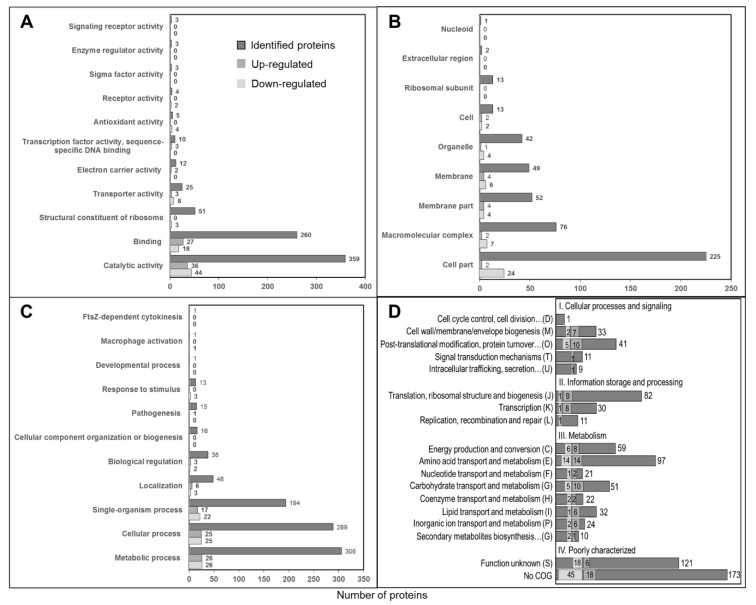
Gene ontology clustering of identified up- and downregulated proteins. (**A**) Molecular Function (MF), (**B**) Cellular Component (CC), (**C**) Biological Process (BP) and (**D**) Cluster of Orthologous Groups (COGs).

**Figure 5 biomolecules-10-00836-f005:**
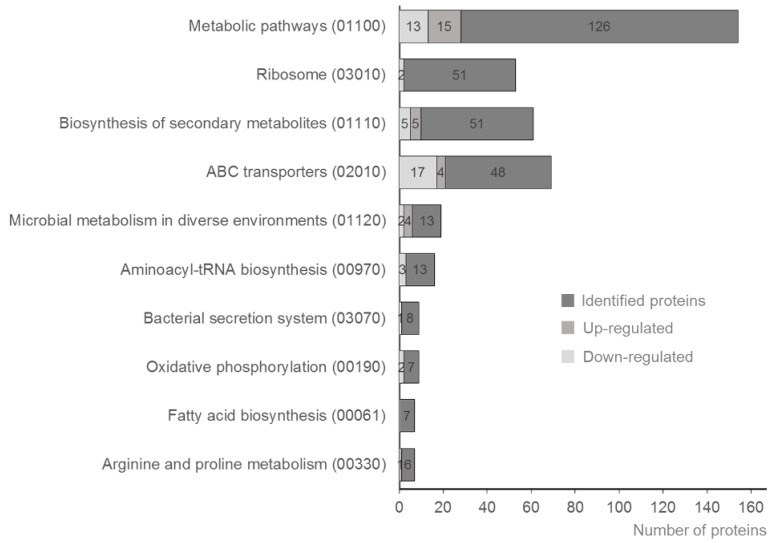
Top 10 KEGG pathways and the distribution of differentially expressed proteins. Number of proteins (*x*-axis) plotted against KEGG pathways (code) on the *y*-axis to show the number of proteins up- and downregulated when *B. abortus* compared to *B. melitensis*.

**Table 1 biomolecules-10-00836-t001:** *Brucella* strains used in this study.

Strain	Designation/ID/Number			Host	Geographical Region
	FAO	ATCC	NCTC		
*B. melitensis* ^T^	16M	23456	10094	Goat	USA
*B. abortus* ^T^	544	23448	10093	Cattle	UK
*B. melitensis*	C *			Sheep	China
*B. abortus*	T *			Cattle	Turkey

^T^ Type strain, FAO—Food and Agriculture Organization * ID assigned to the field strains deposited at the culture collections of Friedrich-Loeffler-Institut, Jena, Germany (C: China, T: Turkey), ATCC—American Type Culture Collection, NCTC—National Collection of Type Cultures, UK.

**Table 2 biomolecules-10-00836-t002:** Number of regulated proteins among type and field strains.

Regulation	Category					
	I M vs. A	II M2 vs. A2	III M2 vs. A	IV M vs. A2	V A2 vs. A	VI M2 vs. M
Down-	173	142	166	223	83	96
Up-	216	199	221	248	97	129

Regulation, down- and up-: downregulated and upregulated, A—*B. abortus*, A2—*B. abortus 544*, M—*B. melitensis* and M2—*B. melitensis* 16M.

**Table 3 biomolecules-10-00836-t003:** KEGG pathways and the distribution of differentially expressed proteins.

KEGG Pathways	Category					
	I	II	III	IV	V	VI
Metabolic pathways		34 ↑				
Carbon metabolism			6 ↑			
Glycolysis/gluconeogenesis				7 ↓		
Pentose and glucuronate interconversions					3 ↓	
Pyruvate metabolism			4 ↑			
2-Oxocarboxylic acid metabolism	4 ↓					
Microbial metabolism in diverse environments		14 ↑	10 ↑			
Biosynthesis of amino acids	11 ↓			14 ↓		
Histidine metabolism	5 ↓	5 ↓	5 ↓	6 ↓		
Valine, leucine and isoleucine biosynthesis				4 ↓		
Purine metabolism		7 ↑				
Pyrimidine metabolism	5 ↑	6 ↑		7 ↑		
Biosynthesis of secondary metabolites	19 ↓			25 ↓		
RNA polymerase				3 ↑		
Ribosome	17 ↑	18 ↓	14 ↓	19 ↑		22 ↓
Bacterial secretion system		5 ↑	5 ↑		4 ↑	5 ↑
ABC transporters			14 ↑			

Number: number of differentially expressed proteins, ↑—upregulated, ↓—downregulated.

**Table 4 biomolecules-10-00836-t004:** Differentially expressed proteins predicted as potentially virulence-associated.

Acc.	Protein Description	Reg	Significance	Reference
D0B8I3*	Antifreeze protein	(+)	Associated with MucR, a transcriptional regulator linked to *Brucella* virulence	[63]
Q8YIA9	Antifreeze protein	(-)		
Q2YM39	Antifreeze protein type I	(-)		
D0B248 *	LipA family protein	(-)		
Q2YIP8	Lipoprotein Omp10	(-)	Reduced virulence in *B. abortus* with gene deletion and used in diagnostics	[64,65,66]
Q2YKV9	Superoxide dismutase [Cu-Zn]	(-)	intracellular survival and used as antigens for subunit vaccines	[19,67]
Q2YRA8	30S ribosomal protein S14	(-)	role *in* cellular adhesion and virulence in *Candida albicans*	[68]
			ribosomal protein L7/L12 based subunit vaccines	[69,70]

Acc. No is the UniProt ID, the protein ID marked with * were moved to UniPrac as they were found to be redundant proteins. Reg—status of protein regulation: (+) denotes upregulation and (-) indicates downregulation of proteins when *B. abortus* is compared with *B. melitensis*.

**Table 5 biomolecules-10-00836-t005:** Differentially expressed proteins among the field isolates in comparison to that of the respective type strains.

UniProt ID	Protein
**Downregulated in Field Isolates**	
Q8YBH7	Bacterial extracellular solute-binding protein family 1
Q8YIX9	Glyceraldehyde-3-phosphate dehydrogenase
Q2YMI0	Phosphatidylserine decarboxylase proenzyme
Q2YLU2	Peptidylprolyl isomerase
Q2YLF8	Leu/Ile/Val-binding protein homolog 1
Q2YLG0	Leu/Ile/Val-binding protein homolog 2
Q2YJA9	Leu/Ile/Val-binding protein homolog 5
Q2YQQ6	Glutelin:Lipoprotein YaeC family:NLPA lipoprotein
Q2YRP7	Ribosome-recycling factor
Q2YR20	Uncharacterized protein
**Upregulated in Field Isolates**	
Q2YJ78	Type IV secretion system protein virB8
Q2YJ79	Type IV secretion system protein virB9
Q2YJ81	Type IV secretion system protein virB10
Q2YJ83	Type IV secretion system-outer membrane lipoprotein
Q2YJ77	Type IV secretion system putative lipoprotein virB7
Q8YB25	Alpha-methylacyl-CoA racemase
Q2YIG7	NADH:flavin oxidoreductase/NADH oxidase
Q2YKS6	Aminotransferase class IV
Q8YCZ2	N-acetylglucosamine-6-phosphate deacetylase
Q8YI04	ATP-dependent DNA ligase
Q2YK32	Catalase
Q2YLX9	Lipoprotein putative
Q2YN45	Probable cytosol aminopeptidase
Q2YRJ0	ATP/GTP-binding site motif A (P-loop)
Q2YRN7	Uncharacterized protein
Q8YBU9	Putative uroporphyrin-iii c-methyltransferase
Q2YKY2	Uncharacterized protein
Q2YL40	Uncharacterized protein
Q2YLM7	Uncharacterized protein
**Downregulated in *B. abortus* and Upregulated in *B. melitensis***	
Q2YME1	Threonylcarbamoyl-AMP synthase
Q2YNW9	Dihydroxy-acid dehydratase
Q8YHA5	Glutaryl-CoA dehydrogenase
Q2YP66	Zinc-containing alcohol dehydrogenase
Q2YQE3	Periplasmic binding protein
Q2YMW1	Uncharacterized protein
Q2YPK6	Uncharacterized protein
**Upregulated in *B. abortus* and Downregulated in *B. melitensis***	
Q2YQV7	50S ribosomal protein L20
Q2YR56	50S ribosomal protein L28
Q2YRY0	ABC-type glycine betaine transport system
Q8YBF5	Maltose-binding periplasmic protein (Sugar ABC transporter)
Q8YCD1	Cystine-binding periplasmic protein
Q8YGE8	Cationic amino acid ABC transporter
Q2YKN7	Uncharacterized protein
Q2YKN9	Uncharacterized protein
Q2YQM2	Uncharacterized protein

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
