# Peer review of "Pan-Proteomic Analysis and Elucidation of Protein Abundance among the Closely Related Brucella Species, Brucella abortus and Brucella melitensis"

_biomolecules, 2020, doi:10.3390/biom10060836_

Round 1

Reviewer 1 Report

This manuscript is of fairly good quality and reads well, presents the research findings clearly.  I would only offer the following suggestions:

(1) In the title, should it be "Pan-proteomic analysis" instead of "Pan-proteomics analysis?"

(2) In the abstract, B. abortus and B. melitensis are the main or major causes of human brucellosis, but not the only ones.  Adding a describing word (like ain or major) would correct this.

(3) In the first line of the introduction, I don't think you need to have "(B.)" because it is common convention that the genus name will be shortened to just the first initial after the first use.

(4) In the same paragraph, it may be useful to include that the classification of Brucella species has not been changed despite the high DNA-DNA identity because of the host preference in livestock and the implications that has in the management of livestock and human zoonotic disease.

(5) Second paragraph of the Introduction: while it is true that Brucella species do not possess plethora of classical virulence factors, I believe the field broadly accepts that the Type IV secretion system (T4SS) is tied to virulence in the species, and thus can be considered a virulence factor.  I am not pushing the authors to say that explicitly (unless they want to), but to at minimum include this in the text as a possibility.

(6) Same paragraph: while it is true that abortus-melitensis cross-protection is challenging I believe the literature of both live and subunit vaccines indicates it is possible to obtain.

(7) M&M, Table 1: It is very important to provide the actual *source* of each of the specific strains used in this study, particularly the type strains because for these strains there are many lab versions throughout the world. Neither the table nor the rest of the Material and Methods explicitly provide this vital information.

(8) Results and discussion, first paragraph: It bears repeating at the start of this section that the study only looked at the proteomes of bacteria that were grown in a rich liquid medium at log phase.  The study did not compare the same strains grown under different conditions.

(9) Line 248: briefly define "razor" here.  Does "razor" mean the same thing as a unique peptide?

(10) Lines 266-7: how does the proteomic clustering recapitulate strain similarities?  That's not clear here.  If comparing to phylogenetic relationships, that should somehow be captured.

(11) Lines 276-8, Table 2, lines 340-1, Table 4, lines 351-2, 354, 361-4, Table 5: The terms "regulated," "up-regulated" and "down-regulated" are used throughout the paper, but what is really being described here is differential protein expression (up or down) in comparisons of species and strains that may or may not have anything to do with actual transcriptional or translational regulation. Instead this could just be different levels of constitutive protein expression in different strains arising from genome sequence variation.  Since you are only comparing different strains grown under the same conditions you cannot know otherwise.

(12) Figure 3: Can delete "times" from the first sentence of the legend because it is redundant.

(13) Line 318: "category" should be "categories."

(14) Lines 337, 340, Table 4 heading, Suppl. table 3: virulence proteins (or virulence-associated proteins) rather than "virulent."  The only virulent proteins I am aware of are prions.

(15) Results and discussion: I am curious again why the T4SS is not considered as a virulence-associated protein here, since it definitely is associated with virulence in Brucella species. VirB of Brucella T4SS is actually a key virulence factor that mediates intracellular survival and is involved in manipulating the mammalian immune response.

(16) Conclusion: Perhaps this section needs to stay brief, but please expand a bit on implications of the species differences you observed. Do the differences you observed in your study give any indications of the difference in biology/phenotype? In your data can you see any preliminary indications of protein differences that might be related to the animal host preference exhibited by the two species (bovine vs. caprine/ovine)? Can you make any speculations that could guide later research? Who would improve it and how would the database be improved? How would the pan-proteome be improved/expanded to provide more utility? I am not suggesting you need to address all of these but more here would be better.

Author Response

Thank you very much for your positive response to our submission. We have taken the comments into consideration in preparing our revision, which has resulted in fine tuning and substantial improvement of the manuscript.

  1. In the title, should it be "Pan-proteomic analysis" instead of "Pan-proteomics analysis?"

Answer: Thank you for this suggestion. We have changed the title as “Pan-proteomic analysis…”

  1. In the abstract, B. abortus and B. melitensis are the main or major causes of human brucellosis, but not the only ones.  Adding a describing word (like main or major) would correct this.

Answer: We agree with the reviewer’s comment and we have rephrased as follows, “The species, B. abortus and B. melitensis, major causative agents of human brucellosis, share remarkably similar genomes, but they differ in their natural hosts, phenotype, antigenic, immunogenic, proteomic and metabolomic properties”.

  1. In the first line of the introduction, I don't think you need to have "(B.)" because it is common convention that the genus name will be shortened to just the first initial after the first use.

Answer: We removed the abbreviation as suggested.

  1. In the same paragraph, it may be useful to include that the classification of Brucella species has not been changed despite the high DNA-DNA identity because of the host preference in livestock and the implications that has in the management of livestock and human zoonotic disease.

Answer: Yes, our statement is to indicate that the classification of Brucella is not changed even after the DNA-DNA hybridization studies, however, the classification is under debate.

  1. Second paragraph of the Introduction: while it is true that Brucella species do not possess plethora of classical virulence factors, I believe the field broadly accepts that the Type IV secretion system (T4SS) is tied to virulence in the species, and thus can be considered a virulence factor.  I am not pushing the authors to say that explicitly (unless they want to), but to at minimum include this in the text as a possibility.

Answer: Thank you very much for this suggestion. We have now included as follows, “Although it is broadly accepted in the field that the Type IV secretion system (T4SS) is some way tied to virulence in Brucella species, Brucella generally lacks classical virulence factors…” (lines 78-80 in the revised manuscript)

  1. Same paragraph: while it is true that abortus-melitensis cross-protection is challenging I believe the literature of both live and subunit vaccines indicates it is possible to obtain.

Answer: The reviewer is correct that cross-protection does occur. We meant that there is a no definitive report on the cross-protection.

  1. M&M, Table 1: It is very important to provide the actual *source* of each of the specific strains used in this study, particularly the type strains because for these strains there are many lab versions throughout the world. Neither the table nor the rest of the Material and Methods explicitly provide this vital information.

Answer: The strains were from culture collection at from the culture collection of grown at the Friedrich-Loeffler-Institut (FLI), Federal Research Institute for Animal Health, Institute of Bacterial Infections and Zoonoses (IBIZ), Jena, Germany. Type strains used in the present study are reference strains distributed by OIE to the reference laboratories. The field isolates used in the present study have been isolated from clinical cases of cattle and sheep brucellosis, respectively. We have included this information in the revised version (lines 91-96) of the manuscript.

  1. Results and discussion, first paragraph: It bears repeating at the start of this section that the study only looked at the proteomes of bacteria that were grown in a rich liquid medium at log phase.  The study did not compare the same strains grown under different conditions.

Answer: Yes, the study is focussed to identify proteome level difference of bacteria grown in rich liquid medium. We don’t want to include several time points at this level as this will complicate.

  1. Line 248: briefly define "razor" here.  Does "razor" mean the same thing as a unique peptide?

Answer: Thanks for pointers. Now we have rephrased accordingly for the clarity, in the identification of 1202 proteins with at least one unique peptide specific for a protein”. (lines 264-265 in the revised manuscript).

A razor peptide is one that can be assigned to more than one protein and is literally a shared peptide. For the quantification of a certain protein from the ion intensities of their belonging peptides this shared peptide is assigned to the protein with the most independent evidence, according to Occam's razor principle and is therefore termed razor peptide. Razor peptides could improve the quantitative info of a protein but usually are not used for its identification.

  1. Lines 266-7: how does the proteomic clustering recapitulate strain similarities?  That's not clear here.  If comparing to phylogenetic relationships, that should somehow be captured.

Answer: Thanks for this pointers. For better clarification on the proteomics clustering, we have rephrased the figure 3 legend as follows, “Figure 3: Protein expression profiling using a label-free quantitative proteomics analysis. A. Heat map analysis of 826 proteins among the four dataset groups and with six cultural replicates per group. The log10 value of the MS signal intensity is shown. Hierarchical clustering of proteins of all samples was performed using Z-score protein intensities for the proteins with P<0.05 and based on elucidation distance. Columns indicate the samples, and rows indicate the proteins. Protein expression values were log2-normalized (label-free quantification (LFQ)) intensities of all proteins quantified across the samples, where red and green indicates high and low intensity, respectively. B, C, D, E, F and G: Volcano plot for category I-VI, respectively. Ratios plotted for log 2-fold change (x-axis) against negative log p-value (y-axis) of the Student's t-test. Each dot represents a protein. (lines 305-314 in the revised manuscript)

  1. Lines 276-8, Table 2, lines 340-1, Table 4, lines 351-2, 354, 361-4, Table 5: The terms "regulated," "up-regulated" and "down-regulated" are used throughout the paper, but what is really being described here is differential protein expression (up or down) in comparisons of species and strains that may or may not have anything to do with actual transcriptional or translational regulation. Instead this could just be different levels of constitutive protein expression in different strains arising from genome sequence variation.  Since you are only comparing different strains grown under the same conditions you cannot know otherwise.

Answer: Yes, we agree with the reviewer on these points. These are only first results on proteome level difference between type and field strains of B. abortus and B. melitensis. The present study may open doors for further understanding of the actual transcriptional or translational regulation through co-culturing of these bacteria with respective host cell lines.

  1. Figure 3: Can delete "times" from the first sentence of the legend because it is redundant.

Answer: Thanks for pointing this out. We have deleted accordingly in the revised draft. (line 306 in the revised version)

  1. Line 318: "category" should be "categories."

Answer: We have corrected this as groups (line 351 in the revised manuscript)

  1. Lines 337, 340, Table 4 heading, Suppl. table 3: virulence proteins (or virulence-associated proteins) rather than "virulent."  The only virulent proteins I am aware of are prions.

Answer: Thanks for the suggestion. We used the term Virulent based on software VirulentPred. Now we have rephrased the term as “potentially virulence-associated…”. (lines 374, 378, table 4 heading at 382 and suppl. Table 3 at line 430  in the revised manuscript)

  1. Results and discussion: I am curious again why the T4SS is not considered as a virulence-associated protein here, since it definitely is associated with virulence in Brucella species. VirB of Brucella T4SS is actually a key virulence factor that mediates intracellular survival and is involved in manipulating the mammalian immune response.

Answer: We agree with the reviewer’s point. And this has already been discussed under the subheading 3.8: Field strains and host adaptability (now Lines 294 to 399 in the revised manuscript). In addition, the following statement is included in the introduction section, “Although it is broadly accepted in the field that the Type IV secretion system (T4SS) is in some way tied to virulence in Brucella species” (Line 78 to 80 in the revised manuscript).

  1. Conclusion: Perhaps this section needs to stay brief, but please expand a bit on implications of the species differences you observed. Do the differences you observed in your study give any indications of the difference in biology/phenotype? In your data can you see any preliminary indications of protein differences that might be related to the animal host preference exhibited by the two species (bovine vs. caprine/ovine)? Can you make any speculations that could guide later research? Who would improve it and how would the database be improved? How would the pan-proteome be improved/expanded to provide more utility? I am not suggesting you need to address all of these but more here would be better.

Answer: Thanks for the suggestions. We have rephrased the conclusion with the following additional sentences, “In conclusion, our quantitative proteomic analysis of reference and field-isolated strains of B. abortus and B. melitensis not surprisingly confirms the existence of proteome level differences between the strains. Besides differences in metabolic pathways, B. abortus and B. melitensis displayed difference in ABC transporters, which were shown to play a role in intracellular survival and virulence of Brucella. Field isolates displayed enhanced abundance in several binding proteins and Type IV secretion system (T4SS), these have been associated with host adaptability and essential pathogenic factors. B. abortus field strain displayed high abundance of ten proteins and seven proteins were of high abundant in B. melitensis. These proteins might be playing role in host specificity. With the exception of seven proteins, all other proteins (n=15) identified as potentially virulence-associated were uncharacterized proteins.” (lines 408-417 in the revised manuscript)

Reviewer 2 Report

The manuscript by Murugaiyan and colleagues describes a large-scale proteomics study of Brucella species. Given the high level of DNA identity and similarity that exists between Brucella species, observed differences in the proteomes could be interesting to identify proteins that may play a role in host specificity. The authors cultured both laboratory and field isolate strains, and a mass spectrometry approach was used to identify and quantify the proteins in each strain.

Overall, the manuscript is composed of several lists of proteins, and it is extremely difficult to decipher what the differences in protein levels actually mean. Moreover, it is difficult to understand what (if any) conclusions the authors have made based on the data.

Specific points:

-What are the conclusions of the study? Yes, there are differences between the strains, but what do those differences mean for the strains? Can a single protein or protein family differentiate between B. abortus and B. melitensis? Can the information be used to assess laboratory vs. field strains? Without some kind of conclusion(s), the manuscript is merely a list of proteins.

-Lines 56-57: The CDC designates Brucella as a "non-tier 1" Select Agent, not "Category B." https://www.selectagents.gov/SelectAgentsandToxinsList.html

-This is a semantic point, but proteins are "produced" rather than "expressed."

Author Response

We would like to thank the reviewers for their careful reading and comments on our manuscript. The following lists our response to each of the reviewer’s suggestions.

Thank you very much for reading our manuscript and for the comments. The manuscript is focused on two different aspects, one availability of reference proteome databases and second the actual proteome level difference between type strains and field isolates of B. abortus and B. melitensis.  We have rephrased the conclusion as follows, “In conclusion, our quantitative proteomic analysis of reference and field-isolated strains of B. abortus and B. melitensis not surprisingly confirms the existence of proteome level differences between the strains. Besides differences in metabolic pathways, B. abortus and B. melitensis displayed difference in ABC transporters, which were shown to play a role in intracellular survival and virulence of Brucella. Field isolates displayed enhanced abundance in several binding proteins and Type IV secretion system (T4SS), these have been associated with host adaptability and essential pathogenic factors. B. abortus field strain displayed high abundance of ten proteins and seven proteins were of high abundant in B. melitensis. These proteins might be playing role in host specificity. With the exception of seven proteins, all other proteins (n=15) identified as potentially virulence-associated were uncharacterized proteins….. (Lines 408-417 in the revised manuscript).

Specific points:

  1. -What are the conclusions of the study? Yes, there are differences between the strains, but what do those differences mean for the strains? Can a single protein or protein family differentiate between abortus and B. melitensis? Can the information be used to assess laboratory vs. field strains? Without some kind of conclusion(s), the manuscript is merely a list of proteins.

Answer: The answer to these questions is not one single protein. We have discussed and listed the list of proteins in the section 3.8. Field strains and host adaptability. At any case, we cannot expect a single protein or a protein family as a factor for host differentiation or species differentiation. We have rephrased the conclusion as mentioned above to include these information’s (Lines 408-417 in the revised manuscript).

  1. -Lines 56-57: The CDC designates Brucella as a "non-tier 1" Select Agent, not "Category B." https://www.selectagents.gov/SelectAgentsandToxinsList.html

Answer: Thank you very much for this suggestion. Now we have rephrased as follows, ´ They are listed ascategory B bioterrorism agents – second highest priority by the Centers for Disease Control and Prevention (https://emergency.cdc.gov/agent/agentlist.asp assessed in March 2020)”.  (lines 56-58 in the revised manuscript)  

  1. -This is a semantic point, but proteins are "produced" rather than "expressed."

Answer: Strictly speaking it might be correct that proteins are synthesized or “produced” rather than expressed, however the terms protein expression and protein expression level are largely used in proteomics research.

Reviewer 3 Report

This is a poorly written and confusing manuscript, describing comparative proteomic analysis of a small set of Brucella strains. While this reviewer is certain that the actual proteomic study was well performed, and has given reliable data, the sample preparation was not rigorous enough and the data has then been poorly presented and over interpreted.  Information is missing to show that samples were prepared from cultures that can be validly compared.  In terms of data analysis it is unfortunate that there is no biology in this manuscript; the data analysis is based on blindly accepting the results from a set of computer predictions with little or no link to the reality of Brucella biology.

Below is a (not exhaustive, non-hierarchical) list of some problems that should be addressed.

1  The first problem is that the proteome analysis uses a set of genome sequence annotations from many studies. These studies have led to different annotations, different methods of identifying protein start and stop sites.  For such a large scale comparison to be valid, this should be made on proteome predictions made by the same annotation pipeline (further manual curation would improve the analysis, but this is understandably too much to ask!).

2  While there are long lists of proteins that the authors claim are detected in different amounts depending on the strain, there are no data available to the reader to see this (without having to trawl through the full data sets submitted on line..).  For key protein hits, the numbers of peptides identified for each protein, the peptide sequences, and the intensities for each protein should be provided in the supplemental data.

3  In general the figure and table legend are uninformative and do not explain the figures. For example for Fig 1, there is no explanation of the arrows. For Fig 2, what do all the numbers in the table mean?

4  From the methods section this reviewer assumes that (lines 86-7)  ‘Each strain was independently cultivated 6 times on Tryptic Soy Broth at 37 °C in the presence of 5% CO2 until mid-logarithmic phase’ represents 6 biological replicates and that the 6 samples were analysed independently.  This should be made clearer. Unfortunately, there is not enough information regarding the culture conditions. Recent clinical isolates can grow more slowly than lab adapted strains. ‘Growth to mid exponential ‘ is very vague, the authors should give the culture volume, type of culture flask, was it shaken, how much, what was final OD, were the numbers of viable bacteria similar?

5  The results/discussion starts with a long section (l 1760199) describing and interpreting old studies that has no relevance to the current study. This should be removed

6  The authors use the terms upregulation and down regulation incorrectly throughout the manuscript.  These terms can only be used to compare the expression of a gene in different conditions for a single strain.  From their data they are measuring the relative abundance of proteins in different strains. This is also dependent on many things other than gene regulation; growth state (see comment above), small genetic differences alter the protein sequence, stability etc.

7  Figure 1 is not needed.

8  As noted above, Fig 2 is difficult to understand, what does the table mean? What does this information bring to the manuscript?

9  The English is very poor in the manuscript.  For example, a genome does not ‘encode an ORF’. An ORF is a DNA sequence which can encode for a protein.  The manuscript continually refers to Virulent proteins. A protein (perhaps with the exception of prions) is not ‘virulent’, it is a factor that can contribute to the virulence of a pathogen.

10  The analysis of ‘virulence factors’ that are identified at different levels is also very basic, with little relevance to what we know about the biology of the organism.  For example, for the ‘antifreeze protein’ they way it is ‘Associated with MucR’ what does associated mean?  Regulated by, co regulated with, interacts with..

Author Response

Thank you very much for the comments and suggestions. The following lists our response to comments

This is a poorly written and confusing manuscript, describing comparative proteomic analysis of a small set of Brucella strains. While this reviewer is certain that the actual proteomic study was well performed, and has given reliable data, the sample preparation was not rigorous enough and the data has then been poorly presented and over interpreted.  Information is missing to show that samples were prepared from cultures that can be validly compared.  In terms of data analysis it is unfortunate that there is no biology in this manuscript; the data analysis is based on blindly accepting the results from a set of computer predictions with little or no link to the reality of Brucella biology.

Answer: Thank you very much for the time spend on reading our manuscript and for the comments. The manuscript is focused on two different aspects, one availability of reference proteome databases and second the actual proteome level difference between type strains and field isolates of B. abortus and B. melitensis. This manuscript also highlights the need for an extended or completely reviewed proteome reference database for better understanding of the host specificity. The computer-based predictions were not taken blindly but defined to open study on testing the actual Brucella biology.

Below is a (not exhaustive, non-hierarchical) list of some problems that should be addressed.

  • The first problem is that the proteome analysis uses a set of genome sequence annotations from many studies. These studies have led to different annotations, different methods of identifying protein start and stop sites.  For such a large scale comparison to be valid, this should be made on proteome predictions made by the same annotation pipeline (further manual curation would improve the analysis, but this is understandably too much to ask!).

Answer: We have utilised the reference proteome database and Gene Ontologies (GO) and other annotations which were common for genes and proteins. Proteomics bioinformatics analysis is not different from that of the analysis of genes.

  • While there are long lists of proteins that the authors claim are detected in different amounts depending on the strain, there are no data available to the reader to see this (without having to trawl through the full data sets submitted on line..).  For key protein hits, the numbers of peptides identified for each protein, the peptide sequences, and the intensities for each protein should be provided in the supplemental data.

Answer: Thanks for the comments. Identification of proteins was based on the detection of more than one unique peptide specific for a protein. This corresponds to the general practice in the field used in the majority of proteomic studies. The raw data and results files with complete details of peptides were submitted to the ProteomXchange EBI-PRIDE repository to be publically and freely available to the readers. It is also mentioned in the abstract as, “All mass spectrometry data are available via ProteomeXchange with identifier PXD006348” (Lines 39-40 in the revised manuscript) and in methods section as follows, “The mass spectrometry proteomics data have been deposited to the ProteomeXchange Consortium [50] via the PRIDE partner repository with the dataset identifier PXD006348” (Lines 182-183 in the revised manuscript).

  • In general the figure and table legend are uninformative and do not explain the figures. For example for Fig 1, there is no explanation of the arrows. For Fig 2, what do all the numbers in the table mean?

Answer:  Thank you very much for this comment. In the revision, we have included the following corrections,

Figure 1 legend is rephrased as suggested, “Figure 1: SDS-PAGE separation of whole cell extract: Lane (a) B. abortus 544, (b) B. abortus T (c) B. melitensis 16 M and (d) B. melitensis C. The strains display comparable bands. Arrows shows distinct bands unique for the isolates with 35 kDa in B. abortus and below 20 kDa in B. melitensis.” (Lines 217 to 219 in the revised manuscript)

Figure 2: The table lists the Brucella species (n=10) included in the B. abortus 2308 pan proteome. Entries: number of isolates; proteins: number of protein entries and percentage: indicates percentage found in the B. abortus 2308 pan proteome. InteractiVenn diagram of Brucella abortus 2308 pan proteome and the entries of five strains (BRUA2- B. abortus (strain 2308) (yellow), BRUAB- B. abortus biovar 1 (strain 9-941) (grey), BRUME- B. melitensis biotype 1 (strain 16M / ATCC 23456 / NCTC 10094) (green), BRUMB- B. melitensis biotype 2 (strain ATCC 23457) (purple) and BRUSU- B. suis biovar 1 (strain 1330)) (blue). (Lines 254 to 260 in the revised manuscript)

  • From the methods section this reviewer assumes that (lines 86-7)  ‘Each strain was independently cultivated 6 times on Tryptic Soy Broth at 37 °C in the presence of 5% CO2 until mid-logarithmic phase’ represents 6 biological replicates and that the 6 samples were analysed independently.  This should be made clearer. Unfortunately, there is not enough information regarding the culture conditions. Recent clinical isolates can grow more slowly than lab adapted strains. ‘Growth to mid exponential ‘ is very vague, the authors should give the culture volume, type of culture flask, was it shaken, how much, what was final OD, were the numbers of viable bacteria similar?

Answer: We have rephrased the method section as follows to include the culture details, “Each strain was independently cultivated 6 times in each 50 ml of on Tryptic Soy Broth at 37 °C in the presence of 5% CO2 with shaking until mid-logarithmic phase. the CFU was around 5x108 cells per ml. The cells were harvested by centrifugation at 11290 g for 5 minutes and after washing twice with phosphate buffer saline, the cells were inactivated and fixed by reconstituting the cell pellets with 300 µl of HPLC grade distilled water and 900 µl of absolute ethanol”. (Lines 92-96 in the revised manuscript)

  • The results/discussion starts with a long section (l 176 to 199) describing and interpreting old studies that has no relevance to the current study. This should be removed

Answer:  Thanks for the suggestion. We believe that this paragraph is important to the readers not so familiar with Brucella and associated earlier reports of proteomics analysis and might be useful for a comprehensive understanding on what has been be reported in the literature.

  • The authors use the terms upregulation and down regulation incorrectly throughout the manuscript.  These terms can only be used to compare the expression of a gene in different conditions for a single strain.  From their data they are measuring the relative abundance of proteins in different strains. This is also dependent on many things other than gene regulation; growth state (see comment above), small genetic differences alter the protein sequence, stability etc.

Answer: We agree with the reviewer that there exists a debate on the use of terms up & down regulation of proteins. However, in majority of proteomics reports, the term up & down regulations are generally used to avoid confusion with the low & high abundance proteins, the terminologies used for blood/serum/plasma/body fluids. Depletion of HAP (high abundance protein) is often used for studies involving above mentioned samples. We also agree with the reviewer on the dependence of the use of these terms on many factors mentioned by the reviewer.

  • Figure 1 is not needed.

Answer: The authors feel that this SDS-PAGE image is important to show the obvious existence of protein level differences in protein bands from whole cell protein extracts of type strains and field strains of genomically similar species of B. abortus and B. melitensis.

  • As noted above, Fig 2 is difficult to understand, what does the table mean? What does this information bring to the manuscript?

Answer: We have rephrased the legend to figure 2 as follows, Figure 2: The table lists the Brucella species (n=10) included in the B. abortus 2308 pan proteome. Entries: number of isolates; proteins: number of protein entries and percentage: indicates percentage found in the B. abortus 2308 pan proteome. InteractiVenn diagram of Brucella abortus 2308 pan proteome and the entries of five strains (BRUA2- B. abortus (strain 2308) (yellow), BRUAB- B. abortus biovar 1 (strain 9-941) (grey), BRUME- B. melitensis biotype 1 (strain 16M / ATCC 23456 / NCTC 10094) (green), BRUMB- B. melitensis biotype 2 (strain ATCC 23457) (purple) and BRUSU- B. suis biovar 1 (strain 1330)) (blue). (lines 254-260 in the revised manuscript)

Figure 2 is important to show the composition of the Brucella Pan Proteome, which is useful for comparing two different species with two or more reference databases.

  • The English is very poor in the manuscript.  For example, a genome does not ‘encode an ORF’. An ORF is a DNA sequence which can encode for a protein.  The manuscript continually refers to Virulent proteins. A protein (perhaps with the exception of prions) is not ‘virulent’, it is a factor that can contribute to the virulence of a pathogen.

Answer: Thanks for the suggestions. Now the revised manuscript as mentioned in the acknowledgement section has been corrected for English grammar by Prof. Marc Howard Rich (native speaker) (lines 443-444 in the revised manuscript)

We have rephrased “encodes ORFs” to “contains ORFs” (line 66 in the revision) and “virulent proteins” (a term used by the software VirulentPred) as “potentially virulence-associated proteins (line 374 in the revised submission).

  • The analysis of ‘virulence factors’ that are identified at different levels is also very basic, with little relevance to what we know about the biology of the organism.  For example, for the ‘antifreeze protein’ they way it is ‘Associated with MucR’ what does associated mean?  Regulated by, co regulated with, interacts with,

Answer: We utilised bioinformatics analysis for identification of potentially virulence associated protein and the literature evidence for the virulence association was presented as in case of MucR. This might be useful for the readers for further research in this direction. It should be noted that among 22 differentially expressed proteins, only seven proteins are defined while remaining 15 proteins were uncharacterised proteins. This underscores the need for further research on these aspects to unravel the actual virulence of the Brucella.

Round 2

Reviewer 2 Report

The authors have made an effort to revise the paper according to the previous comments.

They have clarified the conclusions, and it should be more helpful to the reader at this point.

Please refer to the CDC website that I provided previously and to the one you have provided. The one you provided lists Brucella as a category B, because Brucella starts with the letter B. For example, Yersinia is listed as category Y. I am really trying to be helpful, and I want you to be as accurate as possible with your designations. Please refer to this link: https://www.selectagents.gov/SelectAgentsandToxinsList.html

Regarding expression vs. production, I completely understand that incorrect usage of phrases and words is sometimes commonplace in certain fields, and again, I was only trying to help you be as accurate as possible.

Reviewer 3 Report

Other than having the English improved and making a few minor modifications to the text, the authors have not taken any of this reviewers comments into account. 

The presentation of the actual data is still restricted to the indigestible data sets in the online depository.  These files are not easily decipherable by the average reader of MDPI. While this is the strict minimum,  many expert proteomics lab make the effort to present their data in an easily accessible manner (for an example look at the suplemental data in Maderia et al https://doi.org/10.1016/j.dib.2018.03.030).